# Permeability of Skin-Mimicking Cell Coatings by Polymers of Complex Architecture Based on Polyoxazolines

**Gia Storti [1], Giulia Romano [1], Kristen Gilmore [1], Nicholas Sadowski [1], Andrii Tiiara [2], Igor Luzinov [2]** 
**and Alexander Sidorenko [1,\***

[1]   Department of Chemistry and Biochemistry, Saint Joseph's University, Phildelphia, PA 19131, USA;
      gs20605058@sju.edu (G.S.)
[2]   Department of Materials Science and Engineering, Clemson University, Clemson, SC 29634, USA
[\*]   Correspondence: asidorenko@sju.edu

**Abstract:** In the scope of drug delivery, the transdermal route is desirable because it provides attainable therapeutic concentrations and has minimal systemic side effects. To make the skin a feasible route for the delivery of therapeutic agents, the biggest challenge is overcoming its natural coating. In this paper, we investigate the effect of the architectures (homopolymer vs. block copolymer vs. hybrid block–graft copolymer) of several amphiphilic polymeric derivatives of poly(2-oxazoline) on skin permeability. The block copolymers are composed of a hydrophobic poly(2-oxazoline) block and a hydrophilic PEG block. The hybrid block–graft copolymers are obtained by grafting hydrophobic side chains of polycaprolactone to a poly(2-oxazoline) backbone. We used the commercially available EpiDerm™ by MatTek, composed of human epidermal cells, as a model of human skin. Two parameters of skin permeation are reported: penetration rate and lag time. We hypothesize that the skin permeation characteristics correlate with the critical micelle concentration and particle size of the studied polymers, while both parameters are a function of the complex architectures of the presented macromolecular constructs. While homopolymer poly(2-oxazolines) show the least permeation, the block copolymers demonstrate partial permeation. The hybrid block–graft copolymers exhibited full penetration through the model skin samples.

**Keywords:** hybrid block–graft copolymer; block copolymer; critical micelle concentration; poly(2-oxazolines)

## 1. Introduction

The stratum corneum (SC), the outermost layer of the skin, comprises about twenty micrometers of dead corneocytes and squamous cells embedded in a lipid matrix. This layer controls absorption and acts as an effective barrier against infection and penetration of large therapeutics [1–3]. Methods such as microneedles and chemical enhancers are used to overcome the SC barrier. However, there are preservation issues and the potential for skin irritation when using such approaches [4–6]. Thus, a need for dynamic biocompatible vehicles capable of permeating the SC is of great importance.

The use of amphiphilic polymers for skin penetration has come to the forefront of drug delivery [7–10]. Micelle-forming nanocarriers have been shown to surpass the SC without damaging the tissue while efficiently delivering therapeutics [11,12]. We previously reported on the synthesis of a family of bio-benign amphiphilic poly(2-oxazoline)-based block copolymers (BCPs) and hybrid block–graft copolymers (HCPs) [13]. Poly(2-oxazolines) are a family of emerging innovative biomaterials that demonstrate notable results compared to other polymeric therapeutics [14–16]. Some of these polymers are readily modifiable, permitting amphiphilicity and solubility to be adjusted based on their functionality [17,18].

In this study, we hypothesize that poly(2-oxazoline)-based BCPs and HCPs will demonstrate amphiphilic properties and permeate the epidermis of human skin (Figure 1). HCPs

are macromolecular constructs constituted from a block copolymer (BCP) backbone with side chains grafted from one of the blocks. Poly(2-oxazoline)-based HCPs composed from both hydrophilic (polyethylene glycol, PEG) and hydrophobic (substituted poly(2-oxazoline)) components are of significant interest for skin permeability, as they are expected to be biodegradable and amphiphilic [19,20]. These qualities allow for low surface tension at the oil/water interface; thus, micelle formation may occur [21]. Self-assembly into micelles makes these amphiphilic hybrids useful in the delivery of medication, where the drug can be enclosed in the hydrophobic portion of the polymer matrix [22–24]. Interested readers are referred to a recent review on biodegradable polymeric micelles that summarizes findings on targeted and controlled drug delivery using PEG; polylactic, polyglycolic, and polyglutamic acids; poly(allyl glycidyl ether); poly(amido amine), etc. [25].

**Figure 1.** A generic structure for (A) aminothiophenol- and (B) cysteamine-functionalized homopolymer, block copolymer (BCP), and hybrid block–graft copolymer (HCP).

The most widely used biodegradable polymer for drug delivery is PEG. However, recent studies have highlighted the possibility of polymer buildup in tissues and an influx of anti-PEG antibodies in the immune system [26–28]. Previous work has shown that short sidechain poly(2-oxazolines) may be an improved PEG alternative due to better biocompatibility [29,30]. Thus, the family of bio-benign amphiphilic poly(2-oxazoline)-based BCPs and HCPs provide a promising substitution for PEG in drug delivery and transdermal applications.

The skin permeation experiments were performed on the EpiDerm™ Skin Model by MatTek. The EpiDerm™ tissues are cultured, normal, human-derived epidermal keratinocytes [31] that are commonly used for skin permeability studies [32,33]. Polymers functionalized with aminothiophenol (A) and cysteamine (B) were studied. The compositions of the polymers were established using $^1$H NMR, Size-Exclusion Chromatography and fluorescence of FITC-labelled amino groups [13]; they are summarized in Table 1.

**Table 1.** Characteristics of the polymers tested in this study: weight-averaged molecular weight (Mw), kDa, polydispersity index (PDI), and fractions of hydrophilic (PEG) and hydrophobic (PCL) blocks.

| | Mw, kDa | PDI | Fraction | |
|---|---|---|---|---|
| | | | PEG | PCL |
| Homopolymer A | 3.9 | 1.18 | - | - |
| BCP A | 10.1 | 1.15 | 0.20 | - |
| HCP A | 17.1 | 1.21 | 0.12 | 0.41 |
| Homopolymer B | 10.2 | 1.11 | - | - |
| BCP B | 19.2 | 1.12 | 0.10 | - |
| HCP B | 24.4 | 1.11 | 0.08 | 0.21 |

## 1.1. Experimental Section

### 1.1.1. Materials

Fluorescein isothiocyanate isomer I (FITC, ≥90% (HPLC)), dimethyl sulfoxide (DMSO) (anhydrous, ≥99.9%), ethanol (anhydrous, 99%), diethyl ether (anhydrous, ACS reagent, ≥99%, contains BHT as inhibitor), and Pur-A-Lyzer™ Mega 6000 Da Dialysis Kit were purchased from Sigma-Aldrich and used without purification. EpiDerm™ EPI-200-X samples for skin permeability studies, EPI-100-LLMM-PRF (phenol red-free) media, and skin permeation devices were purchased from MatTek (Ashland, MA, USA) and used without purification.

### 1.1.2. Synthesis of the Hybrid Block–Graft Copolymer

This process is extensively described in our previous manuscript published elsewhere [13], shown in Scheme 1. The HCPs probed for skin permeation were synthesized using a three-step process. The first step was the functionalization of commercially available 2-isopropenyl-2-oxazoline with 4-aminothiophenol or 2-(Boc-amino) ethanethiol (common structure HS-R) via a thiol-ene click chemistry reaction. The second step consisted of ring-opening polymerization of the functionalized oxazoline monomers initiated by either methyl p-toluenesulfonate (leading to homopolymers A or B) or 2.0 kDa PEG methyl ether tosylate (macroinitiator). The latter resulted in BCP A and BCP B. (Note: homopolymer B and BCP B were Boc-deprotected by heating in a vacuum.) The last step was grafting poly-caprolactone (PCL) side chains via ring-opening polymerization initiated by amino groups of (2-oxazoline) monomer units, yielding the amphiphilic biodegradable polyoxazoline-containing HCP A and HCP B.

**Scheme 1.** The three-step synthetic scheme used for the synthesis of the HCPs. (A) aminothiophenol and (B) cysteamine.

### 1.1.3. NMR Spectroscopy

$^1$H NMR spectra were measured on a Bruker Avance III with a 400 MHz frequency using deuterated chloroform CDCl$_3$. All spectra were analyzed using TopSpin software.

### 1.1.4. Fluorescently Labeled Polymers

A solution of fluorescein isothiocyanate (FITC) was added dropwise to the homopolymers, BCPs, and HCPs in DMSO (1:10) overnight at room temperature, Scheme 2. The products were purified by precipitation in a 1:3 ethanol: diethyl ether solution and placed

in dialysis for three days. The polymers were dried under vacuum and analyzed using fluorescence.

**Scheme 2.** Modification of hybrid block–graft copolymer with florescent fluorescein isothiocyanate. (A) aminothiophenol and (B) cysteamine.

### 1.1.5. Fluorescence Spectroscopy

A Cary Eclipse Fluorescence Spectrometer was used to collect the fluorescence spectra. The experiment was performed in emission mode. The PMT detector voltage of 750 V was used. The excitation wavelength used for the negative control, homopolymers, BCPs, and HCP was 490 nm, while the positive control wavelength was set to 260 nm.

### 1.2. Experimental Methods

### 1.2.1. Dynamic Light Scattering

Dispersions of HCP A and HCP B (0.1%) were prepared in water and prepared in a pH 6.86 buffer (0.01%). Each sample was filtered using a 0.45 µm filter then particle size was measured using a Malvern Zetasizer Nano ZS instrument at 30 °C; light source: He-Ne laser with 633 nm wavelength, cuvette cell type: glass cuvette PCS8501 made with quartz glass; the volume of the cuvette: 3.5 mL; light pathlength: 10 mm. Solvent: DI water, $n = 1.330$; viscosity, 0.7920 cP; polymers, $n = 1.47$; material absorption, 0.01.

### 1.2.2. Equilibration of EpiDerm™ Tissues

Under sterile conditions, the tissues were removed from the agarose packaging and placed in a 6-well plate with 0.9 mL of EPI-100-LLMM-X-PRF media. The samples were incubated overnight before use.

### 1.2.3. Skin Penetration Studies

The EpiDerm™ samples were removed from the incubator and placed inside the skin permeation devices provided by MatTek. Next, 0.4 mL of the fluorescently labeled polymer solutions (0.01%) was added to the receiver chamber of the devices (Figure 2). The media in the receiving chamber (5 mL) was collected after desired incubation time and substituted by new media. Fluorescence of the receiver solution was measured immediately after collection. The samples were taken and measured every 1–2 h for up to 24 h. Upon completion of the penetration experiments (24 h), the fluorescence of the donor solution was measured. The difference in fluorescence intensity of the donor solution before and after experimentation corresponded to the cumulative penetration of the polymers through the tissue film. The calculations for cumulative concentration and lag time were made in accordance with the recommendations given by MatTek [30].

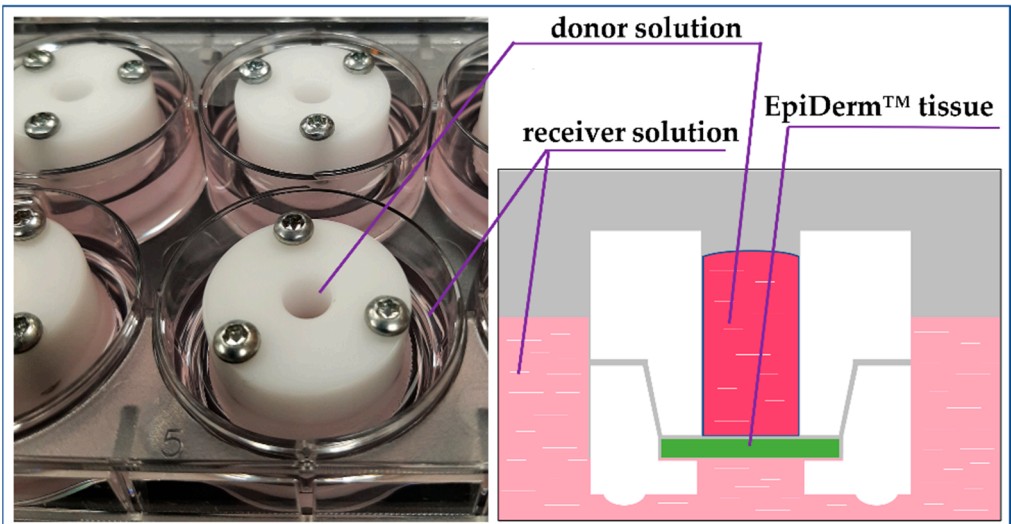

**Figure 2.** Skin penetration studies were performed in MatTek devices (cross-section is shown on the (**right**)) placed in 6-well plates (**left**). Donor solution was placed in the donor chamber on top of the EpiDerm tissue film. The receiver solution was collected for fluorescence measurement after desired incubation time.

### 1.2.4. Determination of Critical Micelle Concentration

The procedure was adapted from our previous work [13]. Aqueous stock solutions of HCPs (0.1%) were prepared. Aliquots of stock HCPs were made using serial dilutions by a factor of two. The absorbance of the HCP as a function of concentration was measured by UV/Vis (600 nm).

## 2. Results and Discussion

The structure of the polymers and HCPs was analyzed using [1]H NMR spectroscopy [13]. Homopolymer A is shown in green, and homopolymer B is shown in red (Figure 3). The long-chain polymer peaks are shown at 'e' and 'd'. The aromatic group of homopolymer A is shown at 6.6 ppm and 7.2 ppm. The Boc-protecting group of homopolymer B is shown at 1.4 ppm.

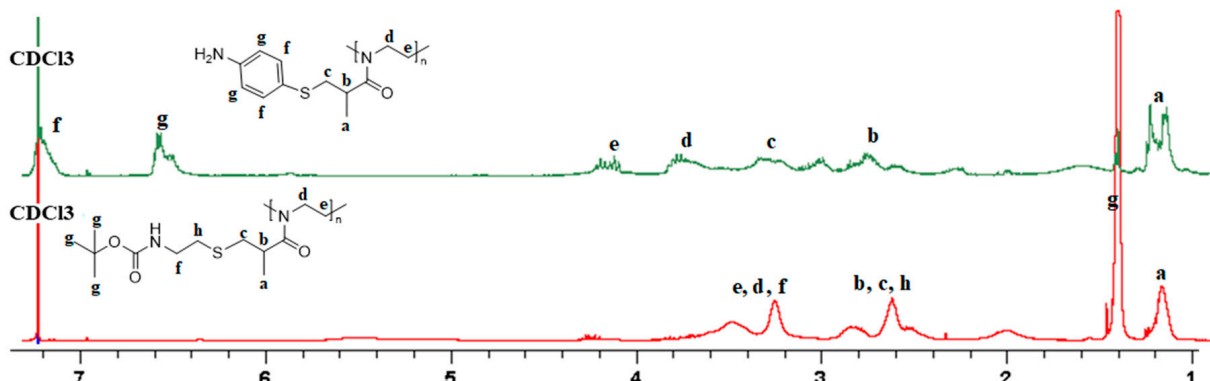

**Figure 3.** [1]H NMR spectra of the homopolymers probed for skin permeation: homopolymer A (green) and homopolymer B (red).

HCP A is shown in green, and HCP B is shown in red (Figure 4). The aromatic group on HCP A (green) remains unchanged and is visible at 6.6 ppm and 7.2 ppm. The PEG block is shown in both spectra at 3.6 ppm. The $\alpha$-carbon on the PCL side chain is visible at 2.33 ppm. The long-chain carbons are shown at 1.67 ppm, 1.40 ppm, and 4.07 ppm.

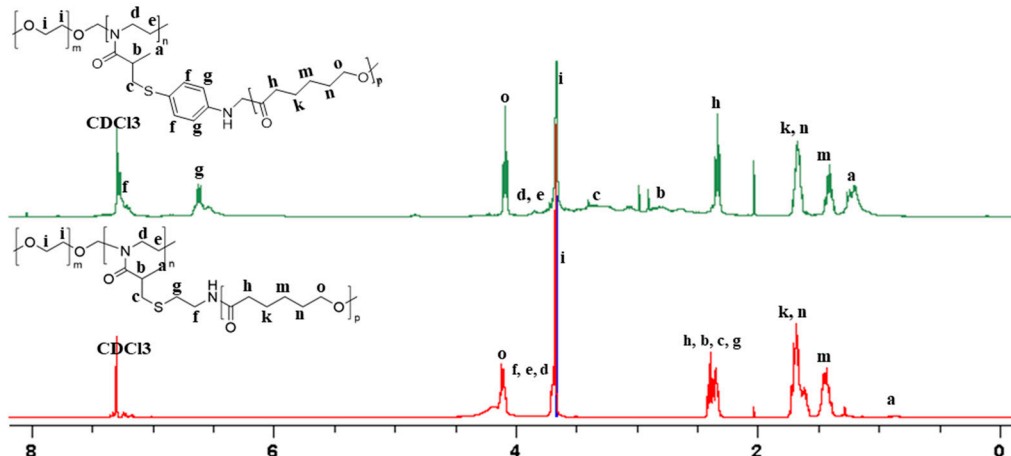

**Figure 4.** ¹H NMR spectra of the HCPs probed for skin permeation: HCP A (green), HCP B (red).

To study the size of the micelles formed by our HCPs, we performed DLS studies (Figures S1–S4) [34,35]. We studied two sets of solutions in DI water and a pH (6.86) buffer. The solutions in the buffer mimic the HCPs in the media when applied to the skin. The summary of our data is reported in Table 2.

**Table 2.** Summary of Micelle Size Collected from Dynamic Light Scattering.

| | Micelle Size, nm | |
| --- | --- | --- |
| | 0.1% in DI Water | 0.01% in pH 6.86 Buffer |
| HCP A | $167.2.2 \pm 71.6$ | $275.5 \pm 81.4$ |
| HCP B | $227.9 \pm 93.2$ | $647.7 \pm 119.2$ |

Three parameters are considered to evaluate permeation efficiency: cumulative concentration, penetration rate, and lag time [36,37]. Cumulative concentration is the total amount of the polymer that penetrates through the sample. The penetration rate is the rate at which the polymer moves through the sample. The lag time is time required to polymer to appear in receiver solution.

The positive control was salicylic acid and the negative control was sodium fluorescein. Both controls were consistent throughout the experiment. Each experiment was performed in triplicate. The neat media solutions showed background fluorescence (intensity of 132 a.u. measured at 490 nm). The fluorescence intensity of the neat media was subtracted from the fluorescence intensity of the receiver media.

After two hours, the positive control fully penetrated the sample skin, and the negative control did not penetrate it, as expected. The fluorescence intensity of the donor solution, the donor solution after 24 h, and the analysis of the cumulative concentration are shown in the supplementary information (Figures S5 and S6).

**Permeation of Aminothiophenol-Functionalized Polymers (A):**

The stock solutions were diluted with media before applying them to the skin sample. The fluorescence intensity of the receiver media was taken every 1–2 h for 24 h. Each experiment was performed in triplicate. The fluorescence intensity of the neat media was subtracted from the fluorescence intensity of the receiver media. The fluorescence intensity of the donor solution, the donor solution after 24 h, and the cumulative concentration are shown in Figures 5 and 6.

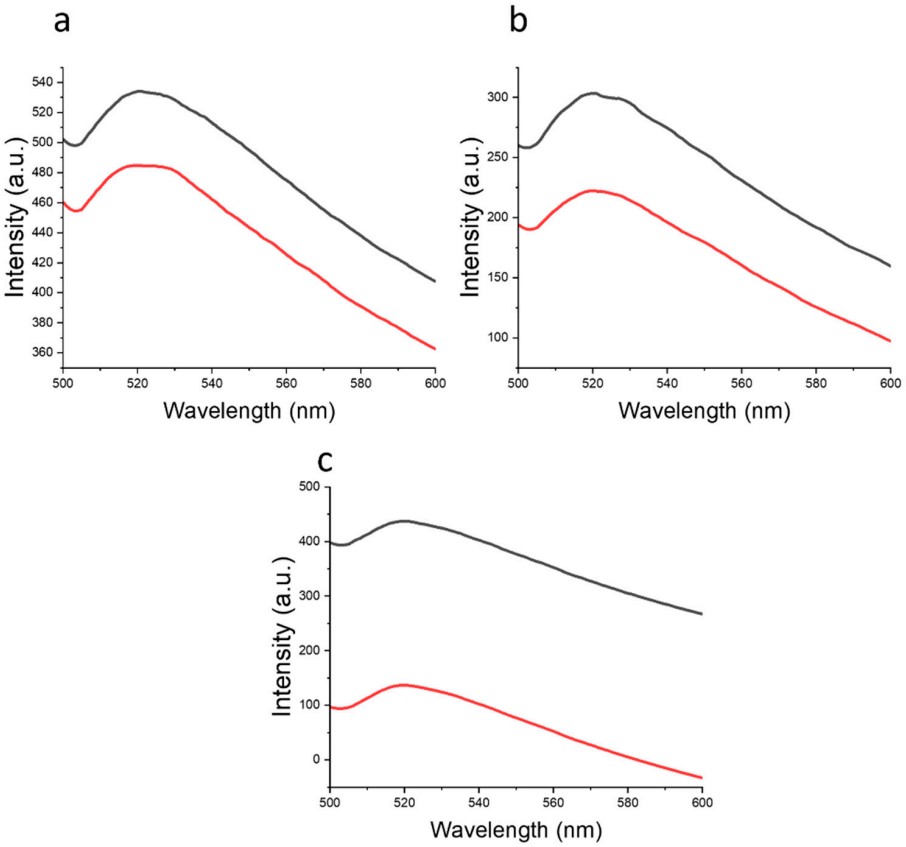

**Figure 5.** Emission spectra of (**a**) donor solution of homopolymer A before the experiment (black) and donor solution after the experiment (red), (**b**) donor solution of BCP A before the experiment (black) and donor solution after (red), and (**c**) donor solution of HCP A before the experiment (black) and donor solution after (red).

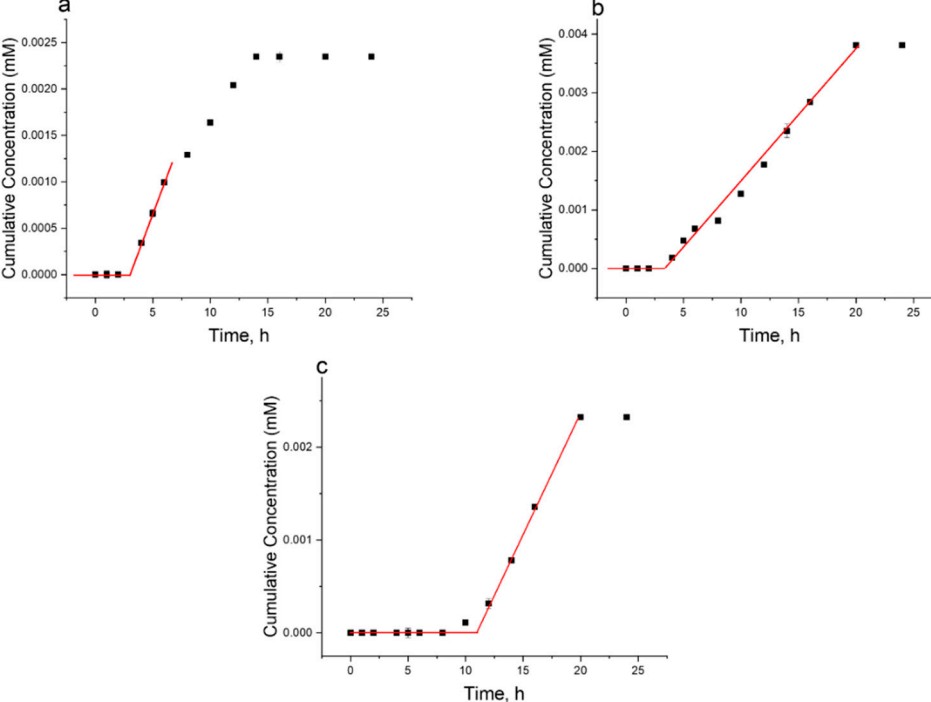

**Figure 6.** Cumulative concentration vs. time of (**a**) homopolymer A, (**b**) BCP A, and (**c**) HCP A.

The cumulative concentration plots show a change in the trend between 5 h and 10 h. We propose that this trend is due to the more rigid non-amphiphilic polymers becoming stuck in the skin sample.

**Permeation of Cysteamine-Functionalized Polymers (B):**

The stock solutions were diluted with media before applying them to the skin sample. The fluorescence intensity of the receiver media was taken every 1–2 h for 24 h. Each experiment was performed in triplicate. The fluorescence intensity of the neat media was subtracted from the fluorescence intensity of the receiver media. The fluorescence intensity and concentration of the donor solution, the donor solution after 24 h, and the cumulative concentration are shown in Figures 7 and 8.

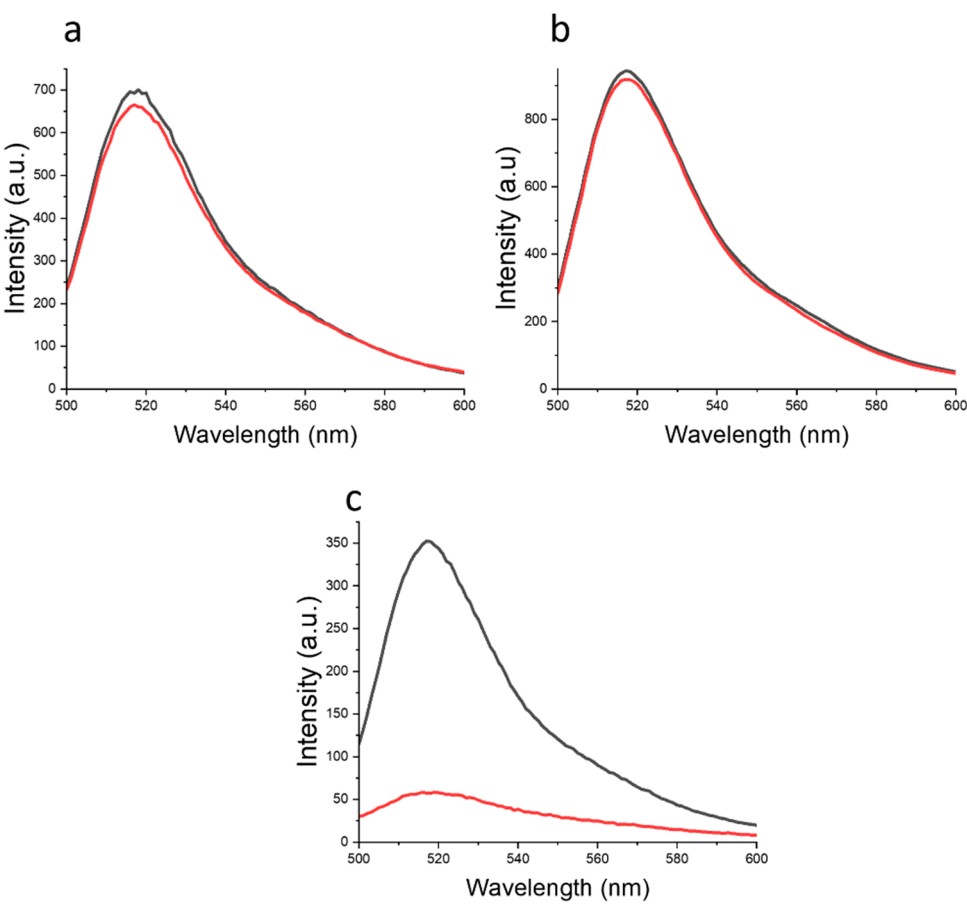

**Figure 7.** Emission spectra of (**a**) donor solution of homopolymer B before the experiment (black) and donor solution after the experiment (red), (**b**) donor solution of BCP B before the experiment (black) and donor solution after (red), (**c**) donor solution of HCP B before the experiment (black) and donor solution after (red).

The average penetration rate and lag time for all samples are shown in Figure 9. Homopolymer A slightly penetrated through the sample skin with a penetration percentage of 8.8%. The lag time of homopolymer A was the shortest of the aminothiophenol polymers. The penetration of homopolymer B was insignificant and could not be measured. BCP A partially penetrated through the sample, with a penetration percentage of 26.8%. The lag time of BCP A was between homopolymer A and HCP A. There was a significant discrepancy in the triplicate data collected for BCP B. We associate this discrepancy with the behavior of the BCP in the cells. Both HCPs fully penetrated through the sample of skin. We observed that the architecture of the HCP is more suitable for skin permeation than the architecture of the homopolymer and BCP. The lag time of HCP B (≤30 min) is much

shorter than HCP A (9.38 h). This is in parallel with the DLS data, as well as previous data collected for their critical micelle concentration (CMC) values [13]. The CMC of HCP A is equal to $1.12 \times 10^{-6}$ M $\pm 6.61 \times 10^{-4}$ and HCP B is equal to $2.48 \times 10^{-7}$ M $\pm 6.28 \times 10^{-9}$ (Figure 10). The recorded micelle size of HCP A is smaller than the micelle size of HCP B (Table 2). We hypothesize that the penetration rate and lag time are not only related to micelle size but also may be related to CMC. The penetration characteristics, micelle size, and CMC depend on the hydrophilic/hydrophobic balance in the HCP.

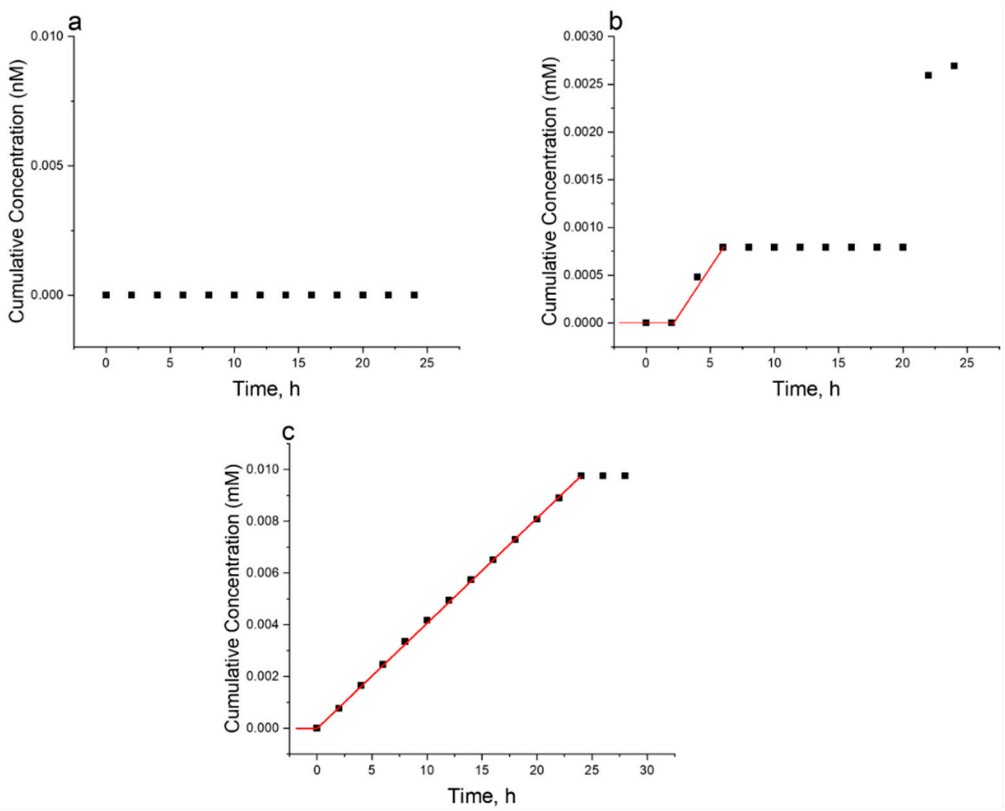

**Figure 8.** Cumulative concentration vs. time of (**a**) homopolymer B (**b**), BCP B, (**c**) HCP B.

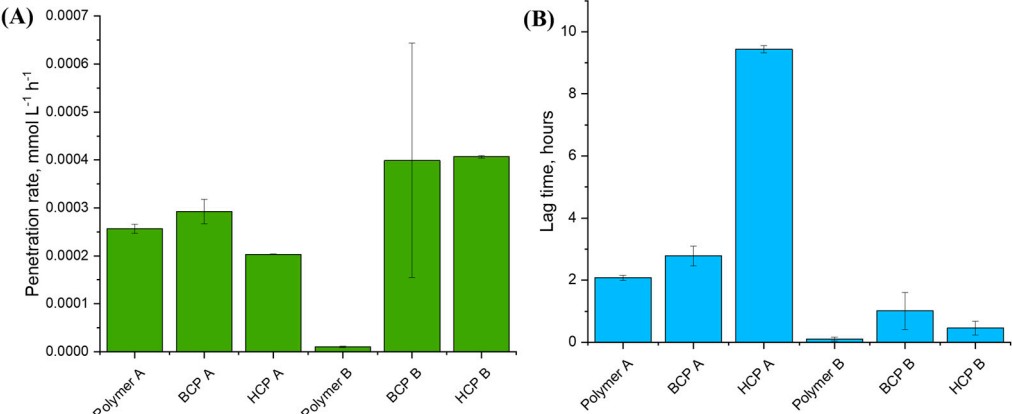

**Figure 9.** Bar graph of averaged (**A**) penetration rate and (**B**) lag time for all studied samples.

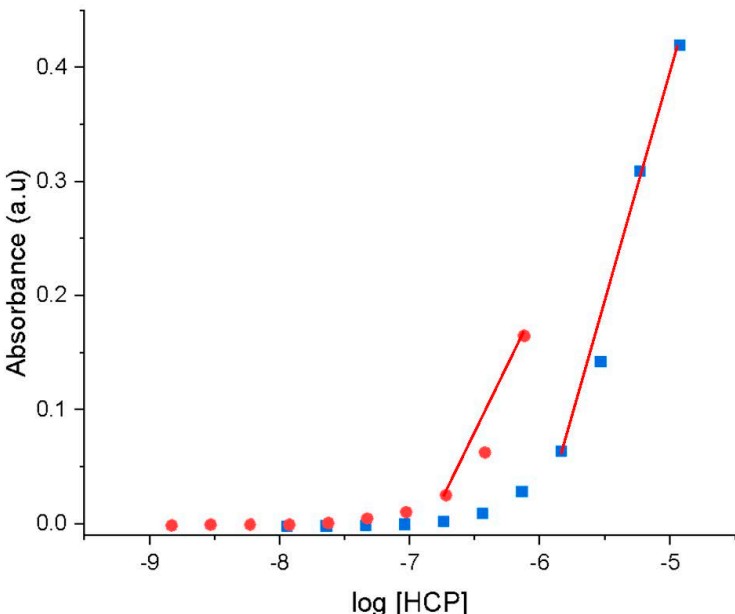

**Figure 10.** Critical micelle concentration of HCP A (blue) and HCP B (red).

### 3. Conclusions

The reported study shows that the hybrid block–graft copolymers based on a poly(2-oxazoline) backbone successfully penetrate the stratum corneum matrix, citing architecture and amphiphilicity as crucial characteristics for penetration efficiency. These hybrid polymers are good alternatives to PEG and can provide a simple and biocompatible vehicle for transdermal drug delivery. The skin permeation of the amino-containing oxazoline homopolymers, block copolymers, and hybrid block–graft copolymers was studied using the EpiDerm™ Skin Model. The homopolymers showed the lowest level of penetration. The block copolymer PEG-block-poly(4-aminothiophenol-2-oxazoline) partially penetrated the skin and had a lag time between that of the corresponding homopolymer and the hybrid polymer. The data collected for PEG-block-poly(cystamine-2-oxazoline) showed a significant discrepancy attributed to the micellar interaction with the cells. The hybrid copolymers showed the most efficient permeation, wherein cysteamine-modified hybrid B penetrated the tissue samples quicker than aminophenol-modified hybrid A. These observations correlate with the critical micelle concentration of the corresponding polymers. Interestingly, the size of the micelles is a less relevant characteristic for skin permeation. We hypothesize that a correct balance of hydrophobicity/hydrophilicity plays a primary role in permeability along with the architecture of the polymer (homopolymer vs. block vs. block–graft copolymer). More data are required to understand the mechanism of polymer permeation, and this is currently under investigation. In particular, more data on the correlation between hybrid polymer composition (fractions of hydrophobic polycaprolactone and hydrophilic PEG) and the length of each block may be of great importance, as well as toxicity studies of poly(2-oxazoline)-based constructs.

**Supplementary Materials:** The following supporting information can be downloaded at: https://www.mdpi.com/article/10.3390/coatings13061007/s1, Figures S1–S4: DLS data for 0.01% HCP A in buffer pH 6.86, 0.1% HCP A in water, 0.01% HCP B in buffer pH 6.86, 0.1% HCP B in water. Figure S5: Emission spectra of (a) donor solution of positive control before experiment (black) and donor solution after the experiment (red) (b) donor solution of negative control before experiment (black) and donor solution after experiment (red). Figure S6: Cumulative concentration vs. time of (a) positive control (b) negative control. Figure S7. Calibration curve of (a) positive control salicylic acid (b) negative control fluorescein sodium. Figure S8. Calibration curve of FITC-aminothiophenol-functionalized (a) homopolymers (b) BCPs (c) HCPs. Figure S9. Calibration curve of FITC-cysteamine-functionalized (a) homopolymers (b) BCPs (c) HCPs.

**Author Contributions:** Conceptualization, A.S. and G.S.; methodology, N.S.; formal analysis, A.T., and I.L.; investigation, G.S., G.R., K.G. and N.S.; data curation, G.R.; writing—original draft preparation, G.S.; writing—review and editing, A.S. and I.L.; supervision, A.S. All authors have read and agreed to the published version of the manuscript.

**Funding:** This research received no external funding.

**Institutional Review Board Statement:** Not applicable.

**Informed Consent Statement:** Not applicable.

**Data Availability Statement:** All data generated or analyzed during this study are included in this published article.

**Conflicts of Interest:** The authors declare no conflict of interest.

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
