# Peer review of "Permeability of Skin-Mimicking Cell Coatings by Polymers of Complex Architecture Based on Polyoxazolines"

_coatings, doi:10.3390/coatings13061007_

Round 1

Reviewer 1 Report

Polyoxazolines and oxazoline spoloimers are promising compounds for use in pharmacology and biomedicine. This is due to their similarity with polypeptides, non-toxicity, biocompatibility, and high synthetic potential. Compared to polyethylene glycol, polyoxazolines are more stable in a physiological environment.

The article Permeability of skin mimicking cell coatings by polymers of complex architecture based on polyoxazolines may be published in Coatings after significant revision.

Notes on the article:

Abstract

1. The abstract needs to be adjusted. You should not refer to other people's research in this section, you should briefly describe the content of your manuscript (what you did, what you got, where you applied or plan to apply).

Introduction

2. Add links where the phrase Manuscript accepted awaiting reference is present.

3. The introduction is written very succinctly. Few references were used for writing. Because of this, it may seem that the topic of polyoxazolines is not relevant and of no interest, which is not true. It is necessary to expand the introduction, provide at least 15 references to modern research. For example, even a query on the MDPI site gives links to interesting works, you can use them (https://doi.org/10.3390/polym13244356, https://doi.org/10.3390/pharmaceutics15030830) and others.

4. What explains the choice of polyoxazolines for the manufacture of micellar carriers? Give a comparative description of micellar carriers based on various polymers - polyethylene glycol, poly-N-vinylpyrrolidone, poly-(ε-caprolactone), polymers and copolymers of glycolic and lactic acids.

Experimental part

Materials and methods

5. It is necessary to give a method for the synthesis of a block copolymer. There is no reference to the work in which the methodology is given.

6. Indicate that deuterochloroform was used for 1H NMR analysis.

Results and its discussion

7. In the region of 7.3 ppm on 1H NMR spectra there is a signal of residual non-deuterated chloroform.

8. Give the characteristics of the obtained homopolymers and copolymers - molecular weight, polydispersity index, solubility in solvents.

9. It is necessary to give a detailed explanation of the emission spectra and graphs of cumulative concentration.

10. Why are HCP hybrid block graft polymers more suitable for application to the skin than BCP homopolymers and block grafts? Justification should be provided.

Conclusions

11. In the conclusion section, do not use abbreviations, but give a full description of the copolymers (for example, instead of HCP A aminothiophenol hybrid-block graft copolymer), which will facilitate the perception of the text. It should indicate which studies are planned in the future. It is desirable to investigate the toxicity of the obtained copolymers in relation to mouse fibroblasts (cell line L929).

English needs some improvement.

Author Response

We appreciate positive evaluation of the manuscript and efforts made by the reviewers in order to improve the quality and clarity of the paper.

In the revised version we addressed all comments and critique expressed by the reviewers.

More specifically:

  1. The abstract needs to be adjusted. You should not refer to other people's research in this section, you should briefly describe the content of your manuscript (what you did, what you got, where you applied or plan to apply).

The abstract has been adjusted. We remove reference to the previous publication and focused on the essential description of the reported study.

  1. Add links where the phrase Manuscript accepted awaiting reference is present.

We brought the reference to unpublished paper in accord with the requirements of the journal.

  1. The introduction is written very succinctly. Few references were used for writing. Because of this, it may seem that the topic of polyoxazolines is not relevant and of no interest, which is not true. It is necessary to expand the introduction, provide at least 15 references to modern research.

Thank you for the suggestions about recent publications. We restructured the intro and added more text relevant to recent studies (including those published in MDPI) of poly(2-oxazoline) polymers. We also added a number of references and believe they now cover the topic adequately.

  1. What explains the choice of polyoxazolines for the manufacture of micellar carriers? Give a comparative description of micellar carriers based on various polymers - polyethylene glycol, poly-N-vinylpyrrolidone, poly-(ε-caprolactone), polymers and copolymers of glycolic and lactic acids.

We also added text rationalizing our choice for polyoxazoline as a backbone polymer with the corresponding references. We also provided the reference to a recent review on drug delivery micelles that describe a number of biodegradable polymers used as such.

  1. It is necessary to give a method for the synthesis of a block copolymer. There is no reference to the work in which the methodology is given.

 By the time of the first submission of current article to Coatings, our paper describing synthesis and characterization of the polymers was under second round review. It is currently available as ASAP at Macromolecules.  In the revised manuscript we refer readers interested to this paper. Yet, we expand the text of current manuscript and provide more details for the synthetic part.

  1. Indicate that deuterochloroform was used for 1H NMR analysis.

 The corresponding text is added.

  1. In the region of 7.3 ppm on 1H NMR spectra there is a signal of residual non-deuterated chloroform.

The region at 7.3 ppm is marked a solvent.

  1. Give the characteristics of the obtained homopolymers and copolymers - molecular weight, polydispersity index, solubility in solvents.

The essential characteristics are added (Table 1) to the revised manuscript.

  1. It is necessary to give a detailed explanation of the emission spectra and graphs of cumulative concentration.

 We modified the text describing the method of measurements (Experimental part). Also, we refer to the vendor’s website that describes methodology of calculations of cumulative concentration and ultimate penetration.

  1. Why are HCP hybrid block graft polymers more suitable for application to the skin than BCP homopolymers and block grafts? Justification should be provided.

HCP is abbreviation for hybrid block-graft copolymers. BCP is different. The BCPs do not form micelles, their hydrophilic-hydrophobic balance is shifted toward hydrophilicity. It looks like HCP of the correct balance (and lower cmc) and architecture (block-graft vs. block vs. homopolymer) demonstrates better permeability. We added more text discussing these aspects in Conclusion.

  1. In the conclusion section, do not use abbreviations, but give a full description of the copolymers (for example, instead of HCP A aminothiophenol hybrid-block graft copolymer), which will facilitate the perception of the text. It should indicate which studies are planned in the future. It is desirable to investigate the toxicity of the obtained copolymers in relation to mouse fibroblasts (cell line L929).

Abbreviations - fixed. More consideration is added to the text of conclusion including future research directions.

Reviewer 2 Report

1. The introduction needs to be rewritten. The importance is not reflrcted.

2. There are many miscellaneous peaks in the NMR. The hybrid-block graft copolymer needs to be purified.

3. Do all amphiphilic polymers have the property?

4. The research mechanism must be added and discussed

Minor editing of English language required.

Author Response

We appreciate positive evaluation of the manuscript and efforts made by the reviewers in order to improve the quality and clarity of the paper.

In the revised version we addressed all comments and critique expressed by the reviewers.

More specifically:

  1. The introduction needs to be rewritten. The importance is not reflrcted.

We restructured the intro and added more text relevant to recent studies (including those published in MDPI) of poly(2-oxazoline) polymers. We also added a number of references and believe they now cover the topic adequately.

  1. There are many miscellaneous peaks in the NMR. The hybrid-block graft copolymer needs to be purified.

The NMR of HCP B is fairly clean. The NMR of HCP A has minor peaks. We observed no difference in NMR of HCP A upon extra precipitation, i.e. it is as pure as it can be. Also, the NMR of BCP A is fairly clean. (See Ref 13). We associate small peaks with side processes related to ROP of caprolactone. We are working to improve the ROP experimental procedure.

  1. Do all amphiphilic polymers have the property?

If I correctly understand the question, the reviewer asks if all amphiphilic polymers demonstrate good skin permeability. We believe that only polymers with the ability to quickly accommodate their conformation to the local environment (polar and/or non-polar) will demonstrate good permeability. Thank you for this question, it is inspiring.

  1. The research mechanism must be added and discussed

We added more details on the method as well as Figure 2 that illustrates the technique.

Reviewer 3 Report

The manuscript entitled “Permeability of skin mimicking cell coatings by polymers of complex architecture based on polyoxazolines” demonstrated  the permeability of a kind of skin mimicking cell coatings. In general, the research concept is timely and of practical interest. The conclusion is well supported by the findings. Therefore, this reviewer would like to recommend the manuscript for publication in Coatings after addressing the following points:

1.         The research concept should be well demonstrated in the “Introduction section”, and the novelty should be highlighted.

2.         Table 1, The units of micellar size should be supplemented.

3.         Error bars should be added to important data discussions.

4.         When discussing the permeation characteristics for the polymers, the present results should be in comparison to that reported in the references, which will make the paper more readable.

The introduction should be well organized, and the English language presentation should be improved.

Author Response

We appreciate positive evaluation of the manuscript and efforts made by the reviewers in order to improve the quality and clarity of the paper.

In the revised version we addressed all comments and critique expressed by the reviewers.

More specifically:

  1. The research concept should be well demonstrated in the “Introduction section”, and the novelty should be highlighted.

We restructured the intro and added more text relevant to recent studies (including those published in MDPI) of poly(2-oxazoline) polymers. We also added a number of references and believe they now cover the topic adequately.

  1. Table 1, The units of micellar size should be supplemented.

The units are added. This is Table 2 now.

  1. Error bars should be added to important data discussions.

Error bars on Figure 9 show standard deviation (of triplicate measurements). We do not show the bars for the single measurements because individual runs deviate from each other. Instead, we treat the runs and obtained their characteristics (penetration rate, lag time) and report with the error bars.  

  1. When discussing the permeation characteristics for the polymers, the present results should be in comparison to that reported in the references, which will make the paper more readable.

The investigated systems (those used in this report as well as other polymer systems) are extremely complex with a number of factors that difficult (if possible, at all) to account. We compare polymers obtain in similar well controlled conditions between themselves to evaluate the effect of architecture and presence the same polymer blocks using the same batch of tissue films. The latter is particularly important as it affects penetration a lot. In the future works, we will try to expand the number of systems and compare them.

Round 2

Reviewer 1 Report

The authors have significantly improved the manuscript. I recommend it for publication.

English is quite acceptable.

Reviewer 2 Report

Accept in present form。

Reviewer 3 Report

The revision work is extensive. Now, this reviewer would like to recommend the manuscript for publication.